# Fast Predictive Uncertainty for Classification with Bayesian Deep Networks

**Marius Hobbhahn**[1]        **Agustinus Kristiadi**[1]        **Philipp Hennig**[1,2]

[1]University of Tübingen
[2]Max-Planck Institute for Intelligent Systems

## Abstract

In Bayesian Deep Learning, distributions over the output of classification neural networks are often approximated by first constructing a Gaussian distribution over the weights, then sampling from it to receive a distribution over the softmax outputs. This is costly. We reconsider old work (Laplace Bridge) to construct a Dirichlet approximation of this softmax output distribution, which yields an analytic map between Gaussian distributions in logit space and Dirichlet distributions (the conjugate prior to the Categorical distribution) in the output space. Importantly, the vanilla Laplace Bridge comes with certain limitations. We analyze those and suggest a simple solution that compares favorably to other commonly used estimates of the softmax-Gaussian integral. We demonstrate that the resulting Dirichlet distribution has multiple advantages, in particular, more efficient computation of the uncertainty estimate and scaling to large datasets and networks like ImageNet and DenseNet. We further demonstrate the usefulness of this Dirichlet approximation by using it to construct a lightweight uncertainty-aware output ranking for ImageNet.

## 1 INTRODUCTION

Quantifying the uncertainty of Neural Networks' (NNs) predictions is important in safety-critical applications such as medical-diagnosis [Begoli et al., 2019] and self-driving vehicles [McAllister et al., 2017, Michelmore et al., 2018], but it is often limited by computational constraints. Architectures for classification tasks produce a probability distribution as their output, constructed by applying the softmax to the point-estimate output of the penultimate layer. However, it has been shown that this distribution is overconfident

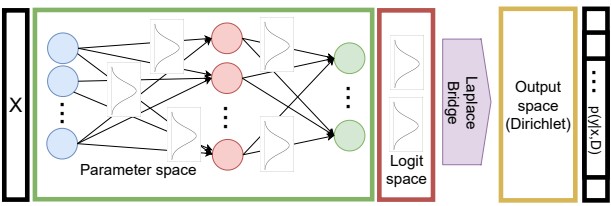

Figure 1: High-level sketch of the Laplace Bridge for BNNs. $p(y|x, D)$ denotes the marginalized softmax output, i.e. the mean of the Dirichlet.

[Nguyen et al., 2015, Hein et al., 2019] and thus cannot be used for predictive uncertainty quantification. Approximate Bayesian methods provide quantified uncertainty over the NN's parameters in a tractable fashion. The commonly used Gaussian approximate posterior [MacKay, 1992a, Graves, 2011, Blundell et al., 2015, Ritter et al., 2018] approximately induces a Gaussian distribution over the logits of a NN [Mackay, 1995], but the associated predictive distribution is not analytic. It is typically approximated by Monte Carlo (MC) integration. This requires multiple samples, making prediction in Bayesian Neural Networks (BNNs) a comparably expensive operation.

Here we reconsider an old but largely overlooked idea originally proposed by David JC MacKay [1998] in a different setting (arguably the inverse of the Deep Learning setting), which transforms a Dirichlet distribution into a Gaussian. When Dirichlet distributions are transformed with the inverse-softmax function, its shape effectively approximates a Gaussian. The inverse of this approximation, which will be called the *Laplace Bridge* (LB) here [Hennig et al., 2012], can also in principle analytically map the parameters of a Gaussian distribution onto those of a Dirichlet distribution. Given a Gaussian distribution over the logits of a NN, one can thus efficiently obtain an approximate Dirichlet distribution over the softmax outputs. However, the bridge was previously used to map in the Gaussian to Dirichlet direction. The inverse direction of the vanilla LB has some limitations, arguably caused by the larger state-space of

*Accepted for the 38th Conference on Uncertainty in Artificial Intelligence* (UAI 2022).

Gaussian relative to the Dirichlet exponential family.

Our contributions are a) We analyze these limits and suggest a solution that allows for the practical application of the LB. b) We show how the result can be used in the context of BNNs (see Figure 1 and 3). c) We empirically evaluate the quality of this approximation, its speed-up, and its performance for out-of-data distribution tasks. d) Finally, we show a use case on ImageNet, leveraging the analytic properties of Dirichlets to improve the popular top-$k$ metric through uncertainties.

## 2 THE LAPLACE BRIDGE

Laplace approximations[1][MacKay, 1992a, Daxberger et al., 2021] are a popular and lightweight method to approximate general probability distributions $q(\mathbf{x})$ with a Gaussian $\mathcal{N}(\mathbf{x}|\boldsymbol{\mu}, \boldsymbol{\Sigma})$ when $q(\mathbf{x})$ is twice differentiable and the Hessian at the mode is positive definite. They set $\boldsymbol{\mu}$ to a mode of $q$, and $\boldsymbol{\Sigma} = -(\nabla^2 \log q(\mathbf{x})|_{\boldsymbol{\mu}})^{-1}$, the inverse Hessian of $\log q$ at that mode. This scheme can work well if the true distribution is unimodal and defined on the real vector space.

The Dirichlet distribution, which has the density function

$$\text{Dir}(\boldsymbol{\pi}|\boldsymbol{\alpha}) := \frac{\Gamma\left(\sum_{k=1}^{K} \alpha_k\right)}{\prod_{k=1}^{K} \Gamma(\alpha_k)} \prod_{k=1}^{K} \pi_k^{\alpha_k - 1}, \quad (1)$$

is defined on the probability simplex and can be "multi-modal" in the sense that the distribution diverges in the $k$-corner of the simplex when $\alpha_k < 1$. This precludes a Laplace approximation, at least in the naïve form described above. However, MacKay [1998] noted that both can be fixed elegantly by a change of variable (Figure 2). Details of the following argument can be found in Appendices B and D. Consider the $K$-dimensional variable $\boldsymbol{\pi} \sim \text{Dir}(\boldsymbol{\pi}|\boldsymbol{\alpha})$ defined as the softmax of $\mathbf{z} \in \mathbb{R}^K$:

$$\pi_k(\mathbf{z}) := \frac{\exp(z_k)}{\sum_{l=1}^{K} \exp(z_l)}, \quad (2)$$

for all $k = 1, \ldots, K$. We will call $\mathbf{z}$ the logit of $\boldsymbol{\pi}$. When expressed as a function of $\mathbf{z}$, the density of the Dirichlet in $\boldsymbol{\pi}$ has to be multiplied by the absolute value of the determinant of the Jacobian

$$\det \frac{\partial \boldsymbol{\pi}}{\partial \mathbf{z}} = \prod_k \pi_k(z_k), \quad (3)$$

---

[1]For clarity: Laplace approximations are *also* one out of several possible ways to construct a Gaussian approximation to the weight posterior of a NN, by constructing a second-order Taylor approximation of the empirical risk at the trained weights. This is *not* the way they are used in this section. The LB is agnostic to how the input Gaussian distribution is constructed as it maps parameters. It could, e.g., also be constructed as a variational approximation, or the moments of Monte Carlo samples.

thus removing the "$-1$" terms in the exponent:

$$\text{Dir}_{\mathbf{z}}(\boldsymbol{\pi}(\mathbf{z})|\boldsymbol{\alpha}) := \frac{\Gamma\left(\sum_{k=1}^{K} \alpha_k\right)}{\prod_{k=1}^{K} \Gamma(\alpha_k)} \prod_{k=1}^{K} \pi_k(\mathbf{z})^{\alpha_k} \quad (4)$$

This density of $\mathbf{z}$, the Dirichlet distribution in the *softmax basis*, can now be accurately approximated by a Gaussian through a Laplace approximation (see Figure 2), yielding an analytic map from the parameter $\boldsymbol{\alpha} \in \mathbb{R}_+^K$ to the parameters of the Gaussian ($\boldsymbol{\mu} \in \mathbb{R}^K$ and symmetric positive definite $\boldsymbol{\Sigma} \in \mathbb{R}^{K \times K}$), given by

$$\mu_k = \log \alpha_k - \frac{1}{K} \sum_{l=1}^{K} \log \alpha_l, \quad (5)$$

$$\Sigma_{k\ell} = \delta_{k\ell} \frac{1}{\alpha_k} - \frac{1}{K} \left[ \frac{1}{\alpha_k} + \frac{1}{\alpha_\ell} - \frac{1}{K} \sum_{u=1}^{K} \frac{1}{\alpha_u} \right]. \quad (6)$$

The corresponding derivations require care because the Gaussian parameter space is evidently larger than that of the Dirichlet and not fully identified by the transformation. A pseudo-inverse of this map was provided as a side result in Hennig et al. [2012]. It maps the Gaussian parameters to those of the Dirichlet as

$$\alpha_k = \frac{1}{\Sigma_{kk}} \left( 1 - \frac{2}{K} + \frac{e^{\mu_k}}{K^2} \sum_{l=1}^{K} e^{-\mu_l} \right) \quad (7)$$

(this equation ignores off-diagonal elements of $\boldsymbol{\Sigma}$, more discussion in Appendix C. Together, Eqs. (5), (6) and (7) will be called the *Laplace Bridge*. For Bayesian Deep Learning, we only use Equation (7) which maps from $\boldsymbol{\mu}, \boldsymbol{\Sigma}$ to $\boldsymbol{\alpha}$. Even though the LB implies a reduction of the distribution's expressiveness, we show in Section 3 that this map is still sufficiently accurate.

## 3 THE LAPLACE BRIDGE FOR BNNS

The Laplace Bridge can be applied to any NN setup that maps from a Gaussian to probabilities by using the softmax. Throughout this paper, we use a last-layer Laplace approximation of the network as successfully used e.g. by Snoek et al. [2015], Kristiadi et al. [2020]. It is given by

$$q(\mathbf{z}|\mathbf{x}) \approx \mathcal{N}(\mathbf{z}|\boldsymbol{\mu}_{\mathbf{W}^{(L)}} \phi(\mathbf{x}), \phi(\mathbf{x})^T \boldsymbol{\Sigma}_{\mathbf{W}^{(L)}} \phi(\mathbf{x})), \quad (8)$$

where $\phi(\mathbf{x})$ denotes the output of the first $L - 1$ layers, $\boldsymbol{\mu}_{\mathbf{W}^{(l)}}$ is the maximum a posteriori (MAP) estimate for the weights of the last layer, and $\boldsymbol{\Sigma}_{\mathbf{W}^{(l)}}$ is the inverse of the negative loss Hessian w.r.t. $\mathbf{W}^{(l)}$, $\boldsymbol{\Sigma}_{\mathbf{W}^{(L)}} = -(\nabla^2_{\mathbf{W}^{(L)}} \mathcal{L})^{-1}$ around the MAP estimate $\mathbf{W}^{(L)}$. Even though last-layer Laplace approximations only use uncertainty from the last linear layer, they empirically perform as well as full Laplace approximations [Kristiadi et al., 2020]. Furthermore, they allow for very fast inference, thus being a good match for

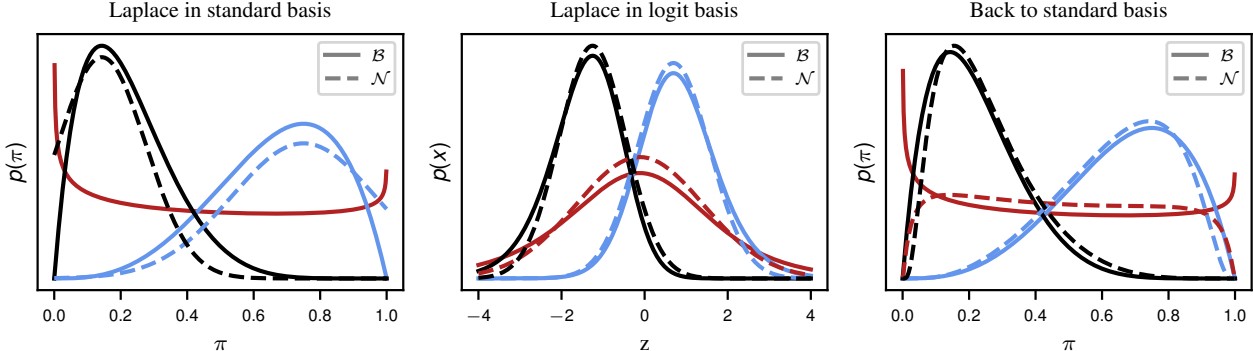

Figure 2: (Adapted from Hennig et al. [2012]). Visualization of the Laplace Bridge for the Beta distribution (1D special case of the Dirichlet) for three sets of parameters. **Left:** "Generic" Laplace approximations of standard Beta distributions by Gaussians. Note that the Beta Distribution (red) does not have a valid approximation because its Hessian is not positive semi-definite. **Middle:** Laplace approximation to the same distributions after basis transformation through the softmax (4). The transformation makes the distributions "more Gaussian" (i.e. uni-modal, bell-shaped, with support on the real line), thus making the Laplace approximation more accurate. **Right:** The same Beta distributions, with the back-transformation of the Laplace approximations from the middle figure to the simplex, yielding an improved approximate distribution. In contrast to the left-most image, the dashed lines now actually are probability densities (they integrate to 1 on the simplex).

the LB. We use diagonal and Kronecker approximations to the Hessian (see Appendix D).

Using the LB we can *analytically* approximate the density of the softmax-Gaussian random variable that is the output of the BNN as a Dirichlet rather than using many samples. As shown in Eq. (7), it requires $\mathcal{O}(K)$ computations to construct the $K$ parameters $\alpha_k$ of the Dirichlet. In contrast, MC-integration has computational costs of $\mathcal{O}(MJ)$, where $M$ is the number of samples and $J$ is the cost of sampling from $q(\mathbf{z}|\mathbf{x})$ (typically $J$ is of order $K^2$ after an initial $\mathcal{O}(K^3)$ operation for a matrix decomposition of the covariance). The MC approximation has the usual sampling error of $\mathcal{O}(1/\sqrt{M})$, while the LB has a fixed but small error (empirical comparison in Section 6.4). This means that computing the LB is faster than drawing a single MC sample while yielding a full distribution.

Further benefits of this approximation arise from the convenient analytical properties of the Dirichlet exponential family. For example, a point estimate of the posterior predictive distribution is directly given by the Dirichlet's mean,

$$\mathbb{E}[\boldsymbol{\pi}] = \left( \frac{\alpha_1}{\sum_{l=1}^{K} \alpha_l}, \dots, \frac{\alpha_K}{\sum_{l=1}^{K} \alpha_l} \right)^{\top}. \qquad (9)$$

This removes the necessity for MC integration and can be computed analytically. Additionally, Dirichlets have Dirichlet marginals: If $p(\boldsymbol{\pi}) = \mathrm{Dir}(\boldsymbol{\pi}|\boldsymbol{\alpha})$, then

$$p\left( \pi_1, \dots, \pi_j, \sum_{k>j} \pi_k \right) = \mathrm{Dir}\left( \alpha_1, \dots, \alpha_j, \sum_{k>j} \alpha_k \right). \qquad (10)$$

Thus marginal distributions of arbitrary subsets of outputs (including binary marginals) can be computed in closed-form.

An additional benefit of the LB for BNNs is that it is more flexible than an MC-integral. If we let $p(\boldsymbol{\pi})$ be the distribution over $\boldsymbol{\pi} := \mathrm{softmax}(\mathbf{z}) := [e^{z_1}/\sum_l e^{z_l}, \dots, e^{z_K}/\sum_l e^{z_l}]^{\top}$, then the MC-integral can be seen as a "point-estimate" of this distribution since it approximates $\mathbb{E}[\boldsymbol{\pi}]$. In contrast, the Dirichlet distribution $\mathrm{Dir}(\boldsymbol{\pi}|\boldsymbol{\alpha})$ approximates the distribution $p(\boldsymbol{\pi})$. Thus, the LB enables tasks that can be done only with a distribution but not a point estimate. For instance, one could ask "what is the distribution of the softmax output of the first $L$ classes?" when one is dealing with $K$-class ($L < K$) classification. Since the marginal distribution can be computed analytically with Eq. (10), the LB provides a convenient yet cheap way of answering this question.

## 4 LIMITATIONS OF THE LAPLACE BRIDGE

There are two limitations to applying the LB as presented in Equation (7). First, the LB assumes that the random variable of the Gaussian sums to zero due to the difference in degrees of freedom between Dirichlet and Gaussian (see Appendix C). Thus, we have to add a correction that projects from any arbitrary Gaussian to one that fulfills this constraint. The resulting Gaussian (see Appendix A) is

$$\mathcal{N}\left( \mathbf{x}|\mu - \frac{\Sigma \mathbf{1}\mathbf{1}^{\top}\mu}{\mathbf{1}^{\top}\Sigma\mathbf{1}}, \Sigma - \frac{\Sigma \mathbf{1}\mathbf{1}^{\top}\Sigma}{\mathbf{1}^{\top}\Sigma\mathbf{1}} \right) \qquad (11)$$

where $\mathbf{1}$ is the one-vector of size $K$.

Second, the softmax-Dirichlet distribution is asymmetric for extremely sparse cases (see Figure 4). These arise in regions where the logistic transform (the 1D special case

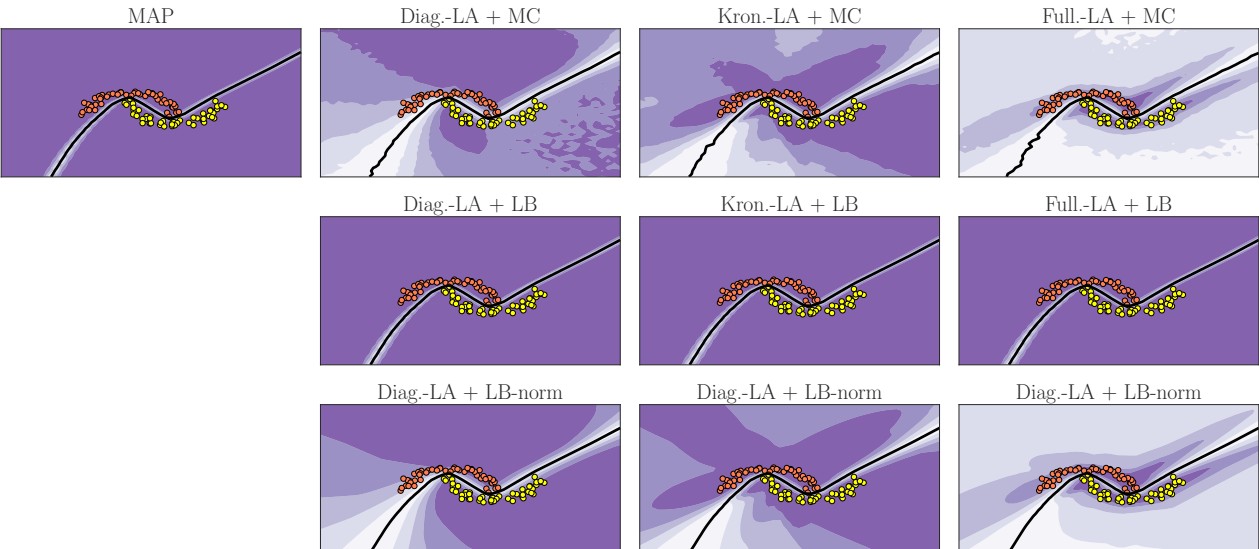

Figure 3: **Left column:** vanilla MAP estimate which is overconfident. **Top row:** mean of softmax applied to Gaussian samples. **Middle row:** mean of the vanilla LB. **Bottom row:** mean of the corrected LB. The vanilla LB yields overconfident prediction far from the data. Our proposed correction fixes this issue, making the LB's approximation close to MC.

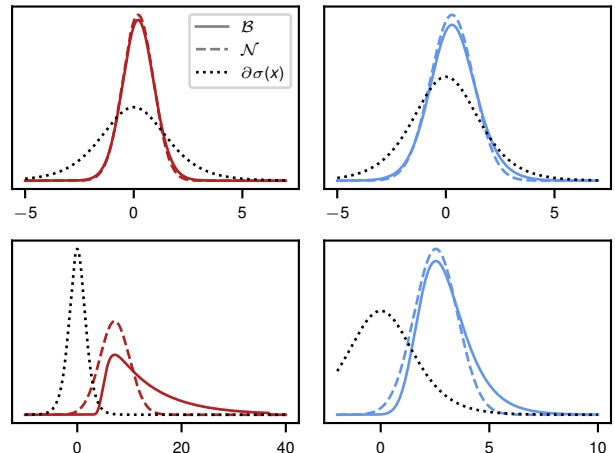

Figure 4: In most scenarios (upper row) the LB provides a good fit. However, in some high-variance scenarios (lower row) the softmax-Dirichlet becomes asymmetric and thus the Gaussian is a suboptimal fit. We propose a correction (right column) that projects the Gaussian into a lower-variance region before applying the LB. This can be understood as "pulling back" the Dirichlet to the dynamic of the logistic function (indicated here by its derivative $\partial\sigma$) and thus yields a better approximation.

of the softmax) is nearly flat (as indicated by its derivative in Figure 4). Therefore, the LA is suboptimal in these high-variance cases.

This limitation can also be explained by looking at Equation (7). We observe that $\Sigma$ contributes linearly to $\alpha$ with

$\frac{1}{\Sigma_{kk}}$ while $\mu$ contributes exponentially with $\exp(\mu_k)$. For settings where $\Sigma$ is small, this doesn't have a large effect. However, when $\Sigma_{kk}$ and $\mu_k$ grow the LB results differ from softmax Gaussian samples. In the LB, the resulting $\alpha$ is dominated by the mean and the linear influence of the variance cannot correct sufficiently. For Monte Carlo sampling, on the other hand, the result is mostly determined by the large variance and then amplified through the softmax. Our proposed normalization to the LB reduces this effect (see Figure 3).

In BNNs, we often encounter such cases, especially far away from the data (see Figure 3 top). Therefore, we propose an additional correction for practical purposes:

$$c = v_{\text{mean}}(\Sigma) \cdot \frac{1}{\sqrt{K/2}} \tag{12}$$

$$\mu' = \frac{\mu}{\sqrt{c}} \tag{13}$$

$$\Sigma' = \frac{\Sigma}{c} \tag{14}$$

where $v_{\text{mean}}(\Sigma)$ denotes the mean variance of $\Sigma$, $v_{\text{mean}}(\Sigma) = \sum_i \Sigma_{ii}$. The factor of $\frac{1}{\sqrt{K/2}}$ is added because we found that higher dimensionalities require less correction. Since our correction is just a rescaling, the zero-sum constrained is still fulfilled. This normalization that can be understood as "pulling back" the distribution into a space where it is symmetric has higher approximation quality. This correction is applied after the zero-sum constraint correction.

We want to point out that our correction is motivated by experimentation and the theoretical insights detailed above.

There is no theoretical derivation from first principles for the correction. We provide additional explanations and figures in Appendix A.

Throughout the paper, we will call this normalizing correction *LB-norm* and explicitly state when we use it. Otherwise, we will use the vanilla version with zero-sum correction.

# 5 RELATED WORK

In BNNs, analytic approximations of posterior predictive distributions have attracted a great deal of research. In the binary classification case, for example, the probit approximation [Gibbs, 1997, Lu et al., 2020] has been proposed already in the 1990s [Spiegelhalter and Lauritzen, 1990, MacKay, 1992b]. However, while there exist some bounds [Titsias, 2016] and approximations of the expected log-sum-exponent function [Ahmed and Xing, 2007, Braun and McAuliffe, 2010], in the multi-class case, obtaining a good analytic approximation of the expected softmax function under a Gaussian measure is an open problem. Our LB can be used to produce a close analytical approximation of this integral. It thus furthers the trend of sampling-free solutions within Bayesian Deep Learning [Wu et al., 2018, Haussmann et al., 2019, etc.]. The crucial difference is that, unlike these methods, the LB approximates the full distribution over the softmax outputs of a deep network.

Previous approaches proposed to model the distribution of softmax outputs of a network directly. Similar to the LB, Malinin and Gales [2018, 2019], Sensoy et al. [2018] proposed to use the Dirichlet distribution to model the posterior predictive for non-Bayesian networks. They further proposed novel training techniques in order to directly learn the Dirichlet. Additionally, different work on Distillation [Malinin et al., 2019, Vadera et al., 2020] takes larger models and distills them into a smaller one. The result of some distillation methods is a Dirichlet similar to the LB. We compare against prior nets in the experiments.

In contrast, the LB tackles the problem of approximating the distribution over the softmax outputs of the ubiquitous Gaussian-approximated BNNs [Graves, 2011, Blundell et al., 2015, Louizos and Welling, 2016, Sun et al., 2017, etc] without any additional training procedure. Therefore the LB can, for example, be used with pre-trained weights on large datasets while prior networks and distillation usually require training from scratch.

# 6 EXPERIMENTS

We conduct multiple experiments. Firstly, we compare the LB to the MC-integral on a 2D toy example (Section 6.1). Secondly, we apply the same comparison to out-of-distribution (OOD) detection in many settings (Section 6.2). Thirdly, we compare the commonly used probit approxima-

tion to the LB in section 6.3 Fourthly, we compare their computational cost and contextualize the speed-up for the prediction process in Section 6.4. Finally, in Section 6.5, we present analysis on ImageNet [Russakovsky et al., 2014] to demonstrate the scalability of the LB and the advantage of having a full Dirichlet distribution over softmax outputs. We extended Laplace torch [Daxberger et al., 2021] for the experiments. Code can be found in the accompanying GitHub repository.[2]

For all experiments, a last-layer Laplace approximation has been applied. This scheme has been successfully used by Snoek et al. [2015], Kristiadi et al. [2020]. We use diagonal and Kronecker-factorized (KFAC)[Ritter et al., 2018, Martens and Grosse, 2015] approximations of the Hessian, since inverting the exact Hessian is too costly. A detailed mathematical explanation and setup of the experiments can be found in Appendix D. While the LB could also be applied to different approximations of a Gaussian posterior predictive such as Variational Inference [Graves, 2011, Blundell et al., 2015], we used a Laplace approximation in our experiments to construct such an approximation. This is for two reasons: (i) it is one of the fastest ways to get a Gaussian posterior predictive and (ii) it can be applied to pre-trained networks which is especially useful for large problems such as ImageNet. Nevertheless, we want to emphasize again that the LB can be applied to any Gaussian over the outputs independent of the way it was generated.

## 6.1 2D TOY EXAMPLE

We train a simple ReLU network on the 2D half-moon problems from scikit-learn [Pedregosa et al., 2011]. As can be seen in Figure 3 the MAP estimate and vanilla LB are overconfident for the reasons discussed in 4 but the normalized version yields a near-perfect fit.

## 6.2 OOD DETECTION

We compare the performance of the LB to the MC-integral (Diagonal and KFAC) on a standard OOD detection benchmark suite, to test whether the LB gives similar results to the MC sampling methods. Following prior literature, we use the standard expected calibration error (ECE) and area under the ROC-curve (AUROC) metrics [Hendrycks and Gimpel, 2016].

For the exact setup, we refer the reader to Appendix D. We use the mean of the Dirichlet to obtain a comparable approximation to the MC-integral. The results are presented in Table 1.

We find that the results of the LB or its normalized version are comparable throughout the entire benchmark suite. Since

---

[2]https://github.com/mariushobbhahn/LB_for_BNNs_official

Table 1: OOD detection results. In all scenarios, the Laplace Bridge (LB) or its normalized version yield comparable results to the MC estimate while being much faster. For MC experiments, we draw 100 samples.

| Train | Test | Diag.-LA + MC | | Diag.-LA + LB | | Diag.-LA + LB-norm | | Kron.-LA + MC | | Kron.-LA + LB | | Kron.-LA + LB-norm | |
|---|---|---|---|---|---|---|---|---|---|---|---|---|---|
| | | ECE ↓ | AUROC ↑ | ECE ↓ | AUROC ↑ | ECE ↓ | AUROC ↑ | ECE↓ | AUROC ↑ | ECE ↓ | AUROC ↑ | ECE ↓ | AUROC ↑ |
| MNIST | FMNIST | **0.464** | 0.975 | 0.478 | **0.981** | 0.498 | 0.951 | 0.390 | 0.987 | 0.553 | 0.977 | **0.364** | **0.990** |
| MNIST | notMNIST | 0.396 | **0.965** | 0.600 | 0.930 | **0.360** | 0.955 | 0.366 | 0.974 | 0.634 | 0.912 | **0.294** | **0.986** |
| MNIST | KMNIST | 0.429 | **0.974** | 0.617 | 0.949 | **0.391** | 0.970 | 0.374 | 0.985 | 0.619 | 0.956 | **0.328** | **0.991** |
| CIFAR10 | CIFAR100 | 0.379 | **0.887** | 0.691 | 0.859 | **0.220** | 0.883 | 0.577 | **0.878** | 0.670 | 0.855 | **0.558** | 0.866 |
| CIFAR10 | SVHN | 0.309 | **0.948** | 0.652 | 0.928 | **0.155** | **0.948** | 0.447 | 0.955 | 0.635 | 0.924 | **0.327** | **0.965** |
| SVHN | CIFAR100 | **0.615** | 0.957 | 0.667 | **0.962** | 0.679 | 0.944 | 0.583 | **0.959** | 0.659 | 0.962 | **0.575** | 0.953 |
| SVHN | CIFAR10 | **0.600** | 0.958 | 0.659 | **0.960** | 0.662 | 0.947 | 0.567 | **0.960** | 0.651 | 0.959 | **0.556** | 0.955 |
| CIFAR100 | CIFAR10 | 0.474 | 0.788 | **0.239** | **0.791** | 0.834 | 0.757 | 0.479 | 0.787 | **0.202** | **0.790** | 0.855 | 0.749 |
| CIFAR100 | SVHN | 0.470 | 0.795 | **0.207** | **0.815** | 0.842 | 0.748 | 0.469 | 0.798 | **0.183** | **0.807** | 0.849 | 0.761 |

Table 2: Comparison of the extended probit approximation with the normalized version of the LB norm. While the probit approximation performs well on in-dist problems, the LB norm is better on out-of-distribution tasks.

| Train | Test | Diag Probit | | | | | Diag LB norm | | | | |
|---|---|---|---|---|---|---|---|---|---|---|---|
| | | MMC ↓ | AUROC ↑ | NLL ↓ | ECE ↓ | Brier ↓ | MMC ↓ | AUROC ↑ | NLL ↓ | ECE ↓ | Brier ↓ |
| MNIST | MNIST | **0.967** | - | 0.050 | 0.024 | 0.002 | 0.944 | - | 0.078 | 0.045 | 0.003 |
| MNIST | FMNIST | 0.597 | **0.971** | 3.827 | 0.523 | 0.128 | **0.589** | 0.951 | **3.538** | **0.498** | **0.124** |
| MNIST | notMNIST | 0.616 | **0.958** | 3.839 | 0.488 | 0.123 | **0.492** | 0.955 | **3.070** | **0.360** | **0.111** |
| MNIST | KMNIST | 0.580 | 0.969 | 4.276 | 0.489 | 0.126 | **0.484** | **0.970** | **3.288** | 0.391 | **0.115** |
| CIFAR10 | CIFAR10 | **0.869** | - | 0.237 | 0.083 | 0.009 | 0.517 | - | 0.727 | 0.433 | 0.029 |
| CIFAR10 | CIFAR100 | 0.589 | 0.882 | 3.334 | 0.485 | 0.123 | **0.319** | **0.883** | **2.590** | **0.220** | **0.099** |
| CIFAR10 | SVHN | 0.510 | 0.946 | 3.097 | 0.394 | 0.114 | **0.273** | **0.948** | **2.457** | **0.155** | **0.094** |

Table 3: Comparison of last-layer vs. full-layer Laplace approximation. Last-layer results are in the upper half and full-layer results are in the bottom half. We find that, as expected, full-layer results are slightly better than for the last-layer approximation.

| Train | Test | Diag.-LA + MC | | Diag.-LA + LB | | Diag.-LA + LB-norm | | Kron.-LA + MC | | Kron.-LA + LB | | Kron.-LA + LB-norm | |
|---|---|---|---|---|---|---|---|---|---|---|---|---|---|
| | | ECE ↓ | AUROC ↑ | ECE ↓ | AUROC ↑ | ECE ↓ | AUROC ↑ | ECE↓ | AUROC ↑ | ECE ↓ | AUROC ↑ | ECE ↓ | AUROC ↑ |
| MNIST | FMNIST | 0.464 | 0.975 | 0.478 | 0.981 | 0.498 | 0.951 | 0.390 | 0.987 | 0.553 | 0.977 | 0.364 | 0.990 |
| MNIST | notMNIST | 0.396 | 0.965 | 0.600 | 0.930 | 0.360 | 0.955 | 0.366 | 0.974 | 0.634 | 0.912 | 0.294 | 0.986 |
| MNIST | KMNIST | 0.429 | 0.974 | 0.617 | 0.949 | 0.391 | 0.970 | 0.374 | 0.985 | 0.619 | 0.956 | 0.328 | 0.991 |
| MNIST | FMNIST | 0.317 | 0.980 | 0.322 | 0.990 | 0.123 | 0.986 | 0.288 | 0.985 | 0.528 | 0.980 | 0.135 | 0.991 |
| MNIST | notMNIST | 0.280 | 0.960 | 0.566 | 0.924 | 0.126 | 0.952 | 0.282 | 0.958 | 0.629 | 0.915 | 0.171 | 0.973 |
| MNIST | KMNIST | 0.309 | 0.976 | 0.557 | 0.955 | 0.112 | 0.972 | 0.279 | 0.981 | 0.615 | 0.958 | 0.152 | 0.986 |

Table 4: Comparison of Prior Networks with the normalized version of the LB norm. PNs consistently outperform the LB. For discussion see main text.

| Train | Test | Prior Network | | | | | Diag LB norm | | | | |
|---|---|---|---|---|---|---|---|---|---|---|---|
| | | MMC ↓ | AUROC ↑ | ECE ↓ | NLL ↓ | Brier ↓ | MMC ↓ | AUROC ↑ | ECE ↓ | NLL ↓ | Brier ↓ |
| MNIST | MNIST | 0.802 | - | 0.184 | 0.246 | 0.008 | **0.944** | - | **0.045** | **0.078** | **0.003** |
| MNIST | FMNIST | **0.273** | **0.995** | **0.212** | 2.659 | **0.098** | 0.589 | 0.951 | 0.498 | 3.538 | 0.124 |
| MNIST | notMNIST | **0.447** | 0.938 | **0.314** | 2.962 | **0.105** | 0.492 | **0.955** | 0.360 | 3.070 | 0.111 |
| MNIST | KMNIST | **0.372** | 0.976 | **0.261** | 3.142 | **0.104** | 0.484 | 0.970 | 0.391 | 3.288 | 0.115 |

the LB is much faster it can be a good replacement for MC in time-sensitive applications.

Furthermore, we compare the LB to prior networks (PNs) in Table 4 since PNs also yield a Dirichlet distribution as an output on classification tasks. We find that PNs outperform the LB in most cases. However, we don't think this is a major problem since they have different aims and use

cases. The LB creates a Dirichlet distribution on top of an already existing Gaussian model while PNs describe a training procedure and have to be trained from scratch. Thus, the primary comparison for the LB should be against sampling and other integral approximations like in Table 2.

Lastly, we compare the LB for a full-layer vs. last-layer Laplace approximation of the network in Table 3. We find

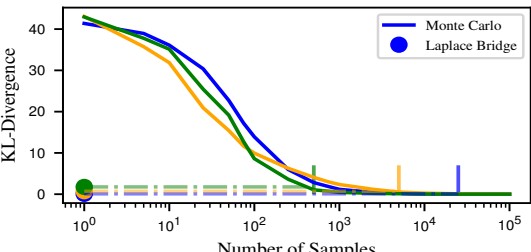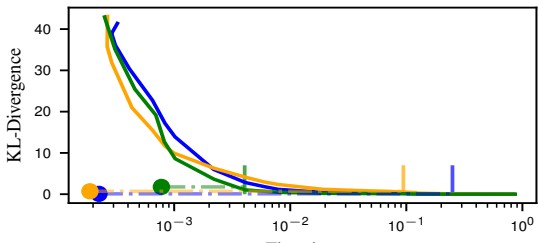

Figure 5: KL-divergence plotted against the number of samples (left) and wall-clock time (right). The Monte Carlo density estimation becomes as good as the LB after around 750 to 10k samples and takes at least 100 times longer. The three lines (blue, yellow, green) represent three different sets of parameters. The short vertical bars indicate where the KL divergence of the samples overtake that of the LB.

that, as expected, the full-layer setting yield slightly better results. However, since the primary advantage of the LB is its speed, we think the natural fit for it is a last-layer approximation.

## 6.3 COMPARISON TO THE PROBIT APPROXIMATION

The multi-class probit approximation [Gibbs, 1997, Lu et al., 2020] is a commonly used approximation for the softmax-Gaussian integral. We compare it to the diagonal normalized LB in Table 2. We find that the LB norm outperforms the probit approximation in most OOD tasks. When we use a KFAC approximation of the Hessian, this trend still holds (see Appendix D).

## 6.4 TIME COMPARISON

We compare the computational cost of the density-estimated $p_{\text{sample}}$ distribution via sampling and the Dirichlet obtained from the LB $p_{\text{LB}}$ for approximating the true $p_{\text{true}}$ over MC-sampling. Different numbers of samples are drawn from the Gaussian, the softmax is applied and the KL-divergence between the histogram of the samples with the true distribution is computed. We use KL-divergences $D_{\text{KL}}(p_{\text{true}} \| p_{\text{sample}})$ and $D_{\text{KL}}(p_{\text{true}} \| p_{\text{LB}})$, respectively, to measure similarity between approximations and ground truth while the number of samples for $p_{\text{sample}}$ is increased exponentially. The true distribution $p_{\text{true}}$ is constructed via MC with 100k samples. The experiment is conducted for three different Gaussian distributions over $\mathbb{R}^3$. Since the softmax applied to a Gaussian does not have an analytic form, the algebraic calculation of the approximation error is not possible and an empirical evaluation via sampling is the best option. The fact that there is no analytic solution is part of the justification for using the LB in the first place.

Figure 5 suggests that the number of samples required such

that the distribution $p_{\text{sample}}$ approximates the true distribution $p_{\text{true}}$ as good as the Dirichlet distribution obtained via the LB is large, i.e. somewhere between 750 and 10k. This translates to a wall-clock time advantage of at least a factor of 100 before sampling becomes competitive in quality with the LB.

To further demonstrate the low compute cost of the LB, we timed different parts of the process for our setup. On our hardware and setup, training a ResNet-18 on CIFAR10 over 130 epochs takes 71 minutes and 30 seconds. Computing a Hessian for the network from the training data can be done with BACKPACK [Dangel et al., 2020] at the cost of one backward pass over the training data or around 29 seconds. This one additional backward pass is the only change to the training procedure compared to conventional training. Since the LB only applies to the last step of the prediction pipeline, it is important to compare it to a forward pass through the rest of the network. Re-using the ResNet-18 and CIFAR10 setup we measure the time in seconds for a forward pass, for the application of the LB, and for the sampling procedure with 10, 100, and 1000 samples. The resulting sum total time for the entire test set is given in Table 5. We find that sampling takes up between 94% (for 1000 samples) and 17% (for 10 samples) of the entire prediction while the LB is only 4%. Thus, the acceleration through the LB is a significant improvement for the prediction process as a whole, not only for a part of the pipeline.

## 6.5 UNCERTAINTY-AWARE OUTPUT RANKING ON IMAGENET

Due to the cost of sampling-based inference, classification on large datasets with many classes, like ImageNet, is rarely done in a Bayesian fashion. Instead, models for such tasks are often compared along a top-$k$ metric (e.g. $k = 5$).

Although widely accepted, this metric has some pathologies: Depending on how close the point predictions are *relative*

Table 5: Contextualization of the timings for the entire predictive process of a ResNet-18 on CIFAR-10. We see that with 1000 samples the forward pass only uses 6% of the time whereas the sampling uses 94%. In contrast the split for the LB is 96% and 4% respectively. We conclude that the LB provides a significant speed-up of the process as a whole.

| # samples in brackets | Forward pass | +MC(1000) | +MC(100) | +MC(10) | +Laplace Bridge |
|---|---|---|---|---|---|
| Time in seconds | $0.300 \pm 0.003$ | $4.712 \pm 0.063$ | $0.488 \pm 0.009$ | $0.059 \pm 0.001$ | $0.013 \pm 0.000$ |
| Fraction of overall time | 0.06/0.38/0.83/0.96 | 0.94 | 0.62 | 0.17 | 0.04 |

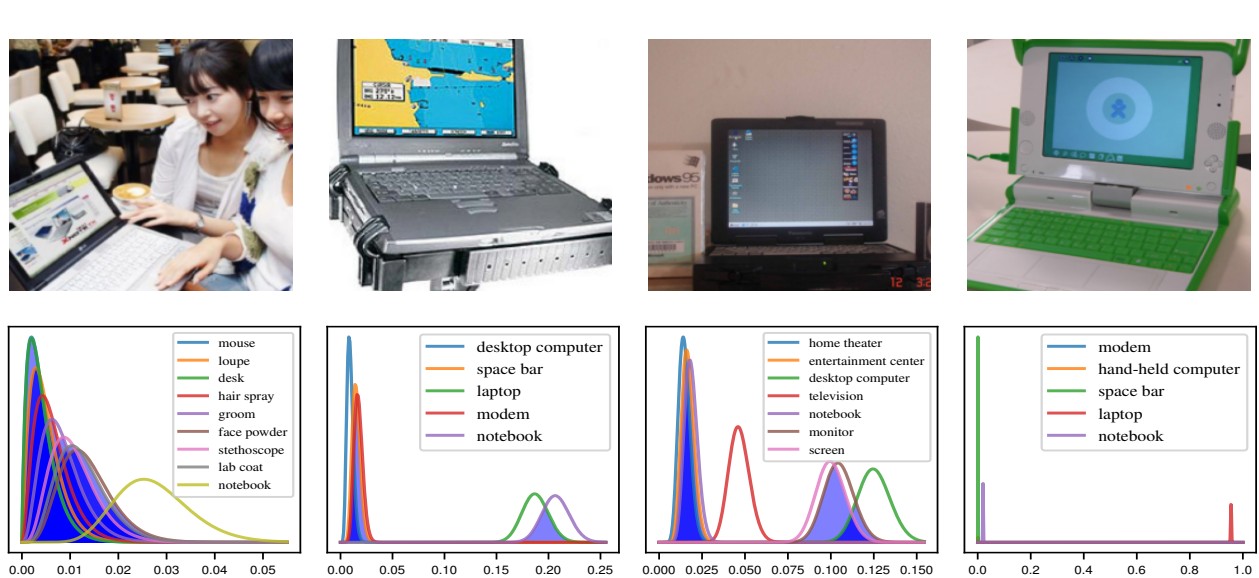

Figure 6: **Upper row:** images from the "laptop" class of ImageNet. **Bottom row:** Beta marginals of the top-$k$ predictions for the respective image. In the first column, the overlap between the marginal of all classes is large, signifying high uncertainty, i.e. the prediction is "do not know". In the second column, "notebook" and "laptop" have confident, yet overlapping marginal densities and therefore yield a top-2 prediction: "either notebook or laptop". In the third column "desktop computer", "screen" and "monitor" have overlapping marginal densities, yielding a top-3 estimate. The last case shows a top-1 estimate: the network is confident that "laptop" is the only correct label.

*to their uncertainty*, the total number of likely class labels should be allowed to vary from case to case. Figure 6 shows examples: In some cases (panel 2) the classifier is quite confident that the image in question belongs to one out of only two classes and all others are highly unlikely. In others (e.g. panel 1), a larger set of hypotheses are all nearly equally probable.

The Laplace Bridge, in conjunction with the last-layer Laplace approximations, can be used to address this issue. To this end, the analytic properties of its Dirichlet prediction are particularly useful: Recall that the marginal distribution $p(\pi_i, \sum_{j \neq i} \pi_j)$ over each component of a Dirichlet relative to all other components is $\mathrm{Beta}(\alpha_i, \sum_{j \neq i} \alpha_j)$.

We leverage this property to propose a simple *uncertainty-aware top-$k$* decision rule inspired by statistical tests. Instead of keeping $k$ fixed, it uses the model's confidence to adapt $k$ (pseudo-code in Algorithm 1).

We begin by sorting the class predictions in order of their expected probability $\alpha_i$. Then we compute the Beta marginal

of the most likely class. Now, we compute the overlap of the next marginal and add that class to the list iff the overlap is more than some threshold (e.g. 0.05). Continuing in this fashion, the algorithm terminates with a finite value $k \leq K$ of "non-separated" top classes.

The intuition behind this rule is that, if any Beta density overlaps with the most likely one more than the threshold of, say, $5\%$, the classifier cannot confidently predict one class over the other. Thus, all classes sufficiently overlapping with the top contender should be returned as the top estimates.

We evaluate this decision rule on the test set of ImageNet. The overlap is calculated through the inverse CDF[3] of the respective Beta marginals. The original top-1 accuracy of DenseNet on ImageNet is $0.744$. In contrast, the uncertainty-aware top-k method yields accuracies of over $0.85$ while average list lengths stay below 3 (see Figure 7). Furthermore, we find that most of the predictions given by the uncertainty-aware metric still yielded a top-1 prediction.

---

[3]Also known as the quantile function or percent point function

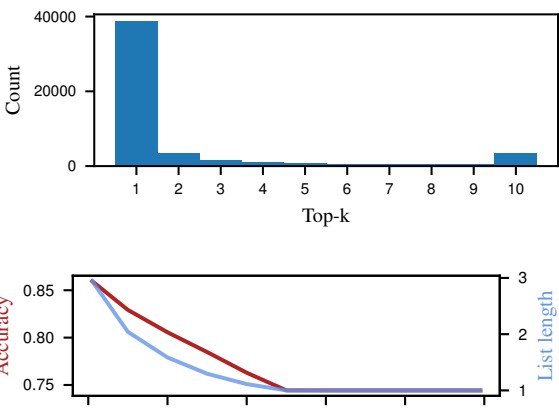

Figure 7: A histogram of ImageNet predictions' length using the proposed uncertainty-aware top-$k$. Results with more than 10 proposed classes have been put into the 10-bin for visibility.

This means that using uncertainty does not imply adding meaningless classes to the prediction. Furthermore, there are non-negligibly many cases where $k$ equals to 2, 3, or 10 (all values larger than 10 are in the 10 bin).

Thus, using the uncertainty-aware prediction rule above, the classifier can use its uncertainty to adaptively return a longer or shorter list of predictions. This not only allows it to improve accuracy over a hard top-1 threshold. Arguably, the ability to vary the size of the predicted set of classes is a practically useful functionality in itself. As Figure 6 shows anecdotally, some of the labels (like "notebook" and "laptop") are semantically so similar to each other that it would seem only natural for the classifier to use them synonymously.

## 7 CONCLUSION

We have adapted a previously developed approximation scheme for new use in Bayesian Deep Learning. Given a Gaussian approximation to the weight-space posterior of a NN (which can be constructed by various means, including another Laplace approximation), and an input, the Laplace Bridge analytically maps the marginal Gaussian prediction on the logits onto a Dirichlet distribution over the softmax vectors. The associated computational cost of $\mathcal{O}(K)$ for $K$-class prediction compares favorably to that of MC sampling. The proposed method empirically preserves predictive uncertainty, offering an attractive, low-cost, high-quality alternative to Monte Carlo sampling. In conjunction with a low-cost, last-layer Bayesian approximation, it is useful in real-time applications wherever uncertainty is required—especially because it drastically reduces the cost of predicting a posterior distribution at test time for a minimal increase

---

**Algorithm 1** Uncertainty-aware top-$k$

**Input:** A Dirichlet parameter $\boldsymbol{\alpha} \in \mathbb{R}^K$ obtained by applying the LB to the Gaussian over the logit of an input, a percentile threshold $T$ e.g. $0.05$, a function class_of that returns the underlying class of a sorted index.

$\tilde{\boldsymbol{\alpha}} = \text{sort\_descending}(\boldsymbol{\alpha})$  // *start with the highest confidence*
$\alpha_0 = \sum_i \alpha_i$
$\mathcal{C} = \{\text{class\_of}(1)\}$  // *initialize top-k, must include at least one class*
$F_1 = \text{Beta}(\tilde{\alpha}_1, \alpha_0 - \tilde{\alpha}_1)$  // *the first marginal CDF*
$l_1 = F_1^{-1}(T/2)$  // *left $\frac{T}{2}$ percentile of the first marginal*
**for** $i = 2, \ldots, K$ **do**
    $F_i = \text{Beta}(\tilde{\alpha}_i, \alpha_0 - \tilde{\alpha}_i)$  // *the current marginal CDF*
    $r_i = F_i^{-1}(1 - T/2)$  // *right $\frac{T}{2}$ percentile of the current marginal*
    **if** $r_i > l_1$ **then**
        $\mathcal{C} = \mathcal{C} \cup \{\text{class\_of}(i)\}$ // *overlap detected, add the current class*
    **else**
        **break**  // *No more overlap, end the algorithm*
    **end if**
**end for**

**Output:** $\mathcal{C}$  // *return the resulting top-k prediction*

in cost at training time. The vanilla LB has some limitations, for which we proposed a simple correction that outperforms alternative softmax-integral approximations such as the commonly used multi-class probit. We demonstrate the utility of the scheme for large-scale Bayesian inference by using it to construct an uncertainty-aware top-$k$ ranking on ImageNet.

## Author Contributions

MH wrote the code, created the figures, and wrote most of the paper. AK gave guidance and supervision and assisted throughout the entire process and greatly helped with the rebuttal. PH had the original idea and provided supervision.

## Acknowledgements

The authors gratefully acknowledge financial support by the European Research Council through ERC StG Action 757275 / PANAMA; the DFG Cluster of Excellence "Machine Learning - New Perspectives for Science", EXC 2064/1, project number 390727645; the German Federal Ministry of Education and Research (BMBF) through the Tübingen AI Center (FKZ: 01IS18039A); and funds from the Ministry of Science, Research and Arts of the State of Baden-Württemberg. MH & AK are grateful to Alexander Meinke for the pre-trained models and the International Max Planck Research School for Intelligent Systems (IMPRS-IS) for support. MH & AK would also like to thank all members of Methods of Machine Learning group for helpful feedback.

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
