# OpenReview forum: "Fast Predictive Uncertainty for Classification with Bayesian Deep Networks"
_auai.org/UAI/2022/Conference — UAI 2022 Poster_

### Official Review · Reviewer_NRTu · 2022-04-09

**Q2(1) Originality/Novelty:** 2
**Q2(2) Significance/Impact:** 3
**Q2(3) Correctness/Technical Quality:** 3
**Q2(6) Clarity Of Writing:** 4
**Q6 Overall Score:** 7
**Q8 Confidence In Your Score:** 4

**Q1 Summary And Contributions:**

The paper considers the probabilistic last layer in neural networks to predict uncertainty. It assumes the Gaussian distribution of the logits and approximates the target distribution for classification after softmaxing with Dirichlet distribution. The paper approximates the relationships between parameters of these distributions with Laplace bridge. The paper demonstrates the drawbacks of Laplace bridge for assymetric Dirichlet distributions and proposes the corrected/normalized version of LB.

**Q2 Assessment Of The Paper:**

More detailed information regarding each of these aspects is given below:

**Q2(4) Quality Of Experiments (Optional):**

4: Excellent: The experimental evaluation is comprehensive and the results are compelling.

**Q2(5) Reproducibility:**

4: Excellent: Key resources (e.g., proofs, code, data) are available and key details (e.g., proof sketches, experimental setup) are comprehensively described for competent researchers to confidently and easily reproduce the main results.

**Q3 Main Strengths:**

* The paper is technically sound and derivations are supported by appendices.
* Novel application of Laplace bridge for uncertainty estimation in neural networks.
* Novel normalizing correction for Laplace bridge.
* Experiments for uncertainty estimation with ImageNet.


**Q4 Main Weakness:**

* Significance is limited, as it is only presented for last-layer probabilities.
* Overall novelty is limited, as it is an application of existing method, Laplace bridge, for existing problem, last-layer Laplace approximation, with key ideas, such as Dirichlet approximation, taken from existing works.

**Q5 Detailed Comments To The Authors:**

Abstract and introduction overemphasize that the method is for Bayesian neural networks in general, as derivations and experiments are only demonstrated for last-layer probabilistic networks and therefore its applicability for more general case is unexplored. Additionally, "This is costly." from the abstract is too short phrase covering lots of existing methods.

Next two technical sections complemented by appendix are reasonably well explained. The main issue for me in the limitation section is that equations 12-14 seem to arise from nowhere, just from experiments. Why exactly do you scale by c, sqrt(c)? How are they related to the dynamic of the logistic function in Figure 4 caption? I believe that analysis of some approximate theoretical functional behavior can support this choice. Another issue is that Figure 4 is confusing for me. If the correction is right column, then why left column contains the dotted derivative plot? Why to apply correction to Beta functions in the right column, if they are already well approximated by Gaussian? Therefore the wording about pulling back does not explain a lot for me. Also, why are they of different colour?

In the related work section, the part about Malinin and Gales [2018, 2019] (I'm not familiar with Sensoy et al. [2018]) is not convincing enough. Is it possible to compare to them, for example in the Cifar10 experiment?

The experimental section contains multiple experiments, including ResNet18/Cifar10 that compares performance according to uncertainty metrics and time with MC where amount of samples is explained. Which exactly MC method is used in this experiment? Are there other methods that lay in between MC and analytical approximation that can trade off quality of uncertainty estimation and time for it? Some other papers also use Brier score for uncertainty quantification, this can complement the ECE.
Another experiment is DenseNet/ImageNet with top-k decision rule where k is adaptive, and it is good to have experiment of that size with results based on uncertainty estimation.

I see that the paper was submitted from ICLR 2021, technical issues were mostly addressed and the material about LB-norm was added.

**Q7 Justification For Your Score:**

Overall I think that the paper is technically sound and experiments are enough to support the claim. On the other side, novelty and significance are moderate, so it is the weak accept score.

UPDATE: After reading the authors' response and other reviews I'm happy to see the paper accepted. I've update the significance score 2->3 and overall score 6->7 to reflect that.

**Q9 Complying With Reviewing Instructions:**

1: Yes.

---

### Official Review · Reviewer_eoxt · 2022-04-11

**Q2(1) Originality/Novelty:** 2
**Q2(2) Significance/Impact:** 2
**Q2(3) Correctness/Technical Quality:** 3
**Q2(6) Clarity Of Writing:** 3
**Q6 Overall Score:** 7
**Q8 Confidence In Your Score:** 4

**Q1 Summary And Contributions:**

The authors revisit the Laplace bridge and use it in reverse and adjusted fashion for mapping a Gaussian over logits in neural classifiers’ penultimate layer, as induced by Gaussian posteriors, to a Dirichlet. The latter represents a (closed-form) distribution over softmax outputs, allowing for inference beyond point estimation without the need for costly posterior samples. Experiments underpin the method’s efficacy and contain a novel, uncertainty-aware form of prediction.

**Q2 Assessment Of The Paper:**

More detailed information regarding each of these aspects is given below:

**Q2(4) Quality Of Experiments (Optional):**

3: Good: The experimental evaluation is adequate, and the results convincingly support the main claims.

**Q2(5) Reproducibility:**

2: Fair: Key resources (e.g., proofs, code, data) are unavailable but key details (e.g., proof sketches, experimental setup) are sufficiently well-described for an expert to confidently reproduce the main results.

**Q3 Main Strengths:**

- The paper proposes a simple, theoretically founded solution to a relevant subproblem of probabilistic inference.
- Its presentation is clear and coherent with good written expression.
- The authors provide a solid mathematical background with proofs and consistent notation.
- The experiments serve to support the paper’s claims and are thoroughly discussed.

**Q4 Main Weakness:**

- Key bits of background information are rather implicit (e.g., the link between Gaussian weight posteriors and Gaussian marginal output distributions (only touched upon in experimental appendix), or the origin of the Laplace bridge’s sum-zero constraint (uniqueness of the mapping?)). This does not hurt the comprehensibility of the main ideas but impedes in-depth understanding.
- The reasoning behind the variance correction is unclear due to the lack of motivation or supporting theory.


**Q5 Detailed Comments To The Authors:**

_Background/explanations_
- There seem to be missing text passages parts, 1) in the motivation for the basis transformation (“Both issues…”, not clear what the second issue is supposed to be), 2) in the motivation for the variance correction (“For these two reasons…”, same problem), and 3) in the derivation of the LB (“this equation ignores off-diagonal elements of Sigma, more discussion below”, I can’t seem to find where this is discussed).
- I’m not sure what is meant by a “completely flat” logistic function, which is by definition isotonic and bounded in [0, 1], nor how the derivative in Figure 4 illustrates this.
- In the variance correction, I take the “mean variance of Sigma” is the mean over diagonal elements – if so, I would recommend to translate this into a formula to avoid misconceptions.
- As mentioned above, the intuition behind the variance correction is not clear to me – in particular, the effect on mu and Sigma apparently depends on whether c is smaller or larger than 1. Some more information should at least be provided in the appendix.
- “Equation (11) is still fulfilled” after variance correction – as Eq 11 is not even technically an equation, it should perhaps rather say “the sum-zero constraint is still fulfilled”.
- The notation does not fully make clear whether the variance correction is applied to the original or sum-zero-corrected moments.
- In Algorithm 1, class_of "returns the underlying class of a sorted index", but takes a scalar (nitpicking, the operation is quite clear).
- In the appendix, it says “Analytically inverting L is done via a lengthy derivation…”, but it would surely not hurt to include it for the sake of completeness.

_Figures_
- The legends in 2 & 4 are not very explicit; 6 lacks axis labels.
- In 2, it might be more consistent to replace x by z in the middle plot. Also, do the blue and black curves actually showcase different scenarios? Lastly, the parameters in the appendix are given “from left to right”, which does not help here.
- In 4, forcing the x axes to the same range might make the effect visible even more clearly.
- In 5, showing the KL divergence on log scale might improve legibility. Also, it does not say what the vertical bars are.

_Potential mistakes_
- Eq 3 seems to be missing an index k in z.
- After Eq 7, in “which maps to…”, the symbols should be bold.
- In Eq 8, the last-layer parameters should probably be indexed with capital L. Also, the definition of the negative loss Hessian should be at the location of the MAP estimator.
- Eq 39 has a k in the first factor that seems to belong in the index.
- Eq 43 has a factor M that is unaccounted for.
- In table 2, the MMC results for in-distribution are highlighted in the Probit column although LB performs better.


**Q7 Justification For Your Score:**

Theoretically founded solution to a relevant subproblem of probabilistic inference with clear presentation and supported by experiments.

**Q9 Complying With Reviewing Instructions:**

1: Yes.

---

### Official Review · Reviewer_sDF1 · 2022-04-12

**Q2(1) Originality/Novelty:** 3
**Q2(2) Significance/Impact:** 3
**Q2(3) Correctness/Technical Quality:** 2
**Q2(6) Clarity Of Writing:** 2
**Q6 Overall Score:** 4
**Q8 Confidence In Your Score:** 2

**Q1 Summary And Contributions:**

This manuscript proposes to approximate the probabilistic prediction in a classification Bayesian NN with Laplace Bridge. This approach avoids drawing samples from the posterior distribution of the weights, while demonstrating comparable results.

**Q2 Assessment Of The Paper:**

More detailed information regarding each of these aspects is given below:

**Q2(4) Quality Of Experiments (Optional):**

2: Fair: The experimental evaluation is weak: important baselines are missing, or the results do not adequately support the main claims.

**Q2(5) Reproducibility:**

2: Fair: Key resources (e.g., proofs, code, data) are unavailable but key details (e.g., proof sketches, experimental setup) are sufficiently well-described for an expert to confidently reproduce the main results.

**Q3 Main Strengths:**

The idea of applying Laplace Bridge is an interesting one and the authors did a good job explaining their choice. The paper is by and large well structured and well written.

**Q4 Main Weakness:**

I find motivating the proposed method via over- or under-confident predictions in NN problematic. For details please refer to Q5.

**Q5 Detailed Comments To The Authors:**

This issue discussed in e.g. [Hein et al. 2019, Guo et al. 2017] is referring to the unreliable quantification of aleatoric uncertainty. The Bayesian NN, however, quantifies the epistemic uncertainty and is not guaranteed to yield better calibrated predictions. But the possibility still exists that a Bayesian NN may be less overfitted due to the prior regularization.
What the proposed approach does is to express the epistemic uncertainty of the last layer in a Bayesian NN in a much more efficient way, without explicitly modifying the aleatoric uncertainty. To this end, the ECE in the experimental part might not be an appropriate choice. Furthermore, I would also be interested in an experiment to compare the epistemic uncertainty in terms of the last layer against that in terms of the entire layer. If the last layer is enough to represent the entire network in terms of the uncertainty, a natural baseline approach would be to train only the last layer with Bayesian weights. And at inference one would only need perform multiple forward passes through the last layer.

**Q7 Justification For Your Score:**

My score is mostly based on my comment in Q5. In summa, I find the idea itself is very interesting but the authors should differentiate epistemic and aleatoric uncertainty in this context. Furthermore, taking only the last layer into account may require more justification. To be better self contained, the authors may want to spend a few sentences on the final conclusion from [Kristdiadi et al. 2020].

**Q9 Complying With Reviewing Instructions:**

1: Yes.

---

### Official Review · Reviewer_iGHZ · 2022-04-12

**Q2(1) Originality/Novelty:** 2
**Q2(2) Significance/Impact:** 2
**Q2(3) Correctness/Technical Quality:** 3
**Q2(6) Clarity Of Writing:** 2
**Q6 Overall Score:** 6
**Q8 Confidence In Your Score:** 4

**Q1 Summary And Contributions:**

This paper looks at using Laplace Bridge method for modeling the output distribution of Bayesian neural networks as Dirichlet. More specifically, the authors propose using last layer Laplace approximation and use Laplace Bridge to compute the output Dirichlet parameters. The authors provide experimental results on combinations of open source datasets and deep neural network models on tasks of OOD detection and uncertainty aware ranking.

**Q2 Assessment Of The Paper:**

More detailed information regarding each of these aspects is given below:

**Q2(4) Quality Of Experiments (Optional):**

2: Fair: The experimental evaluation is weak: important baselines are missing, or the results do not adequately support the main claims.

**Q2(5) Reproducibility:**

1: Poor: Key details (e.g., proof sketches, experimental setup) are incomplete/unclear, or key resources (e.g., proofs, code, data) are unavailable.

**Q3 Main Strengths:**

The Laplace bridge method helps get an output distribution over last-layer Laplace approximation using a single forward pass, and thus saves the high computation requirement needed in Monte Carlo integral. The paper also addresses a critical pitfall in straightforward LB, and presents a correction that helps in better uncertainty estimation.

**Q4 Main Weakness:**

While the approach seems technically sound, there are certain details that are hard to capture in the paper. Furthermore, the authors haven't provided any training/hyperparameter details, nor have they provided their implementation. Finally, it seems that a motivation behind using LB is to reduce the run-time latency for uncertainty estimation applications (using last-layer Laplace approximation). There exists a set of distillation methods that look at this problem that should be discussed and compared against [1, 2]. Since these methods can work with any arbitrary ensembles, it would be interesting how they fare against compressing last layer Laplace approximation.

References

[1] Malinin, Andrey, Bruno Mlodozeniec, and Mark Gales. "Ensemble distribution distillation." ICLR (2020).

[2] Vadera, Meet, Brian Jalaian, and Benjamin Marlin. "Generalized bayesian posterior expectation distillation for deep neural networks." UAI (2020).

**Q5 Detailed Comments To The Authors:**

- The first line of the abstract says, "In Bayesian Deep Learning, distributions over the output of classification neural networks are approximated by first constructing a Gaussian distribution over the weights". This isn't true generally.

- In section 4, is says "First, the LB assumes that the random variable of the Gaussian sums to zero (Appendix C)". I've looked at Appendix C, but I am still unable to see this. Could you elaborate on what random variables are you talking about, and how do they sum to zero?

- Fig 6, I get how the Beta distributions are formed, but I am not sure how to exactly interpret the overlaps. Could the authors explain what they mean? When predicting top-k estimate, shouldn't we just compute the predictive probabilities for all classes and pick the top-k from there?

- There's existing work that looks at other approximate Bayesian inference techniques and different measures of uncertainty (along with the distillation versions) that can be extracted for OOD detection [1-4]. I believe this should be discussed, and the authors should compare their results with some other approximate inference techniques. For a paper like this, I think it's fine if the method doesn't outperform other approximate inference techniques, since the novelty lies in the method itself, but comparing against distillation methods for OOD detection would be very uesful.

References

[1] Vadera, Meet P., Adam D. Cobb, Brian Jalaian, and Benjamin M. Marlin. "Ursabench: Comprehensive benchmarking of approximate bayesian inference methods for deep neural networks." arXiv preprint arXiv:2007.04466 (2020).

[2] Malinin, Andrey, Bruno Mlodozeniec, and Mark Gales. "Ensemble distribution distillation." ICLR (2020).

[3] Vadera, Meet, Brian Jalaian, and Benjamin Marlin. "Generalized bayesian posterior expectation distillation for deep neural networks." UAI (2020).

[4] Depeweg, S., Hernandez-Lobato, J., Doshi-Velez, F. & Udluft, S. Decomposition of Uncertainty in Bayesian Deep Learning for Efficient and Risk-sensitive Learning. ICML (2018).



**Q7 Justification For Your Score:**

I am rating this paper as borderline reject as it's missing some critical experimental comparisons, and there's lack of details on the Reproducibility front.

**Q9 Complying With Reviewing Instructions:**

1: Yes.

---

### Decision · Program_Chairs · 2022-05-15

**Decision:**

Accept (Poster)

**Comment:**

Meta Review: Reviewers were primarily concerned about lack of novelty, as the paper provides a relatively incremental extension to the Laplace bridge, which itself is an application of methods from MacKay. On balance, reviewers were mostly satisfied with author responses, and felt the paper could be providing a useful contribution. It would certainly help frame the paper better if it had comparisons with (and at least discussions of) simple fast baselines for uncertainty in deep learning, not based on Laplace, such as SWAG, https://proceedings.neurips.cc/paper/2019/file/118921efba23fc329e6560b27861f0c2-Paper.pdf.